# Enhancing the Antifungal Activity and Ophthalmic Transport of Fluconazole from PEGylated Polycaprolactone Loaded Nanoparticles

**DOI:** 10.3390/polym15010209

**Published:** 2022-12-31

**Authors:** Alshaimaa M. Almehmady, Khalid M. El-Say, Manal A. Mubarak, Haneen A. Alghamdi, Njood A. Somali, Alaa Sirwi, Rahmah Algarni, Tarek A. Ahmed

**Affiliations:** 1Department of Pharmaceutics, Faculty of Pharmacy, King Abdulaziz University, Jeddah 21589, Saudi Arabia; 2Department of Natural Products and Alternative Medicine, Faculty of Pharmacy, King Abdulaziz University, Jeddah 21589, Saudi Arabia; 3Pharmaceutical Care Department, King Abdulaziz University Hospital, Jeddah 21589, Saudi Arabia

**Keywords:** fluconazole, poly (ε-caprolactone), nanoparticles, antifungal, hydrogel, ophthalmic treatment, industrial development

## Abstract

Fungal eye infections are caused mainly by an eye injury and can result in serious eye damage. Fluconazole (FLZ), a broad-spectrum antifungal agent, is a poorly soluble drug with a risk of hepatotoxicity. This work aimed to investigate the antifungal activity, ocular irritation, and transport of FLZ-loaded poly (ε-caprolactone) nanoparticles using a rabbit eye model. Three formulation factors affecting the nanoparticle’s size, zeta potential, and entrapment efficiency were optimized utilizing the Box-Behnken design. Morphological characteristics and antifungal activity of the optimized nanoparticles were studied. The optimized nanoparticles were loaded into thermosensitive in situ hydrogel and hydroxypropylmethylcellulose (HPMC) hydrogel ophthalmic formulations. The rheological behavior, in vitro release and in vivo corneal transport were investigated. Results revealed that the percentage of poly (ε-caprolactone) in the nanoparticle matrix, polymer addition rate, and mixing speed significantly affected the particle size, zeta potential, and entrapment efficiency. The optimized nanoparticles were spherical in shape and show an average size of 145 nm, a zeta potential of −28.23 mV, and a FLZ entrapment efficiency of 98.2%. The antifungal activity of FLZ-loaded nanoparticles was significantly higher than the pure drug. The developed ophthalmic formulations exhibited a pseudoplastic flow, prolonged the drug release and were found to be non-irritating to the cornea. The prepared FLZ pegylated nanoparticles were able to reach the posterior eye segment without eye irritation. As a result, the developed thermosensitive in situ hydrogel formulation loaded with FLZ polymeric nanoparticles is a promising drug delivery strategy for treating deep fungal eye infections.

## 1. Introduction

Scientists and pharmacologists are facing many challenges with the process of ocular drug delivery. These difficulties are attributed to the unique anatomy and physiology of the eye. Lacrimation, reflex blinking, tear-film turnover or nasolacrimal duct drainage might cause quick elimination of the drug from the eye [1]. The anterior eye segment is easily accessible for ocular drug delivery, hence topical eye drop instillation is the preferred technique. Delivering the drug to the posterior eye segment might be challenging due to the short drug residence on the eye, ocular barriers and limited drug bioavailability at the site of action [2]. An available choice for ocular drug delivery is the administration of the drug via direct injection into the eye. This procedure has many drawbacks, including expensive costs, low patient compliance, and an increased risk of injection-related complications [3]. Another choice is the implementation of ocular implants for sustained drug delivery to the posterior of the eye. However, it requires many injections or surgery, with associated risks [3]. 

Nanotechnology-based ocular therapy, particularly those produced from biodegradable polymers, are currently gaining high interest and can be an option to overcome current ocular therapy limitations [4]. Various forms of nanoparticles have been investigated for ocular delivery, such as liposomes, nanotransferosomes, and polymeric nanoparticles. Polymeric nanoparticles have shown great potential in the controlled delivery of various drugs with enhanced bioavailability to the ocular sites of action [4]. Ocular drug delivery based on particulate systems were reported to have the ability for potential particle entrapment in the eye’s mucin, which leads to a prolonged residence and slower drainage. Moreover, sustained drug release and improved drug absorption in case of drug-loaded nanoparticles were reported to enhance bioavailability [5]. Varshochian et al. have applied the PLGA/PVA polymeric system to prepare bevacizumab-loaded polymeric nanoparticles as a potential therapy for ocular neovascularization. Bevacizumab is effective in the treatment of ocular neovascularization. However, it has a short residence in the vitreous humor, which requires frequent administration via intravitreal injections [6]. Interestingly, bevacizumab-loaded polymeric nanoparticles showed a sustained release and a drug concentration in the vitreous humor that reached more than 500 ng/mL which is the minimum drug concentration that stops the vascular endothelial growth factor-induced angiogenesis for around two months. The ocular delivery of dexamethasone, an efficient anti-inflammatory drug, was also studied following the development of drug-loaded nanoparticles. Ryu et al. formulated dexamethasone-loaded nanoparticles incorporated in an alginate matrix. This formulation was later prepared into dry tablets, and administered with a specific preocular applicator and resulted in an enhancement in the ocular drug bioavailability by 2.6-fold compared to a commercial pure drug eyedrops [7]. Diclofenac-loaded cationic chitosan grafted methoxy poly(ethylene glycol)-poly(ε-caprolactone) nanoparticles were developed by Shi et al. The amphiphilic property of the studied polymer enables self-assembly to form nanoparticles while its positive charges interact with the mucin’s negative charges to extend nanoparticle retention time at the sites of action. Comparing this formulation to commercial diclofenac eye drops, it showed no toxicity and improved permeability and retention of the drug (1.4-fold higher). The amount of diclofenac in the rabbits’ aqueous humor was 2.3 times more than what was seen when the commercial medication ocular formulation was used [7]. The choice of polymers for ocular delivery is usually based on their biocompatibility and biodegradability. Poly (ε-caprolactone) (PCL) is a biocompatible and biodegradable synthetic polymer that was approved by the US Food and Drug Administration (FDA). Due to its low toxicity and slower degradation rate, PCL has been widely used for ophthalmic controlled drug delivery [8,9]. PCL has been applied as intraocular controlled release implants to deliver dexamethasone. PCL has been fabricated into a release microfilm to deliver triamcinolone acetonide to prevent proliferative vitreoretinopathy after intraocular surgery [9]. 

Antifungal agents mainly belong to the following categories: polyenes (such as amphotericin B, natamycin and nystatin), azoles/imidazoles (such as ketoconazole, miconazole and econazole), triazoles (such as itraconazole, voriconazole, posaconazole, fluconazole and ravuconazole) and echinocandins (such as caspofungin, micafungin, anidulafungin) [10]. Fluconazole (FLZ), a synthetic triazole derivative, possesses a broad-spectrum antifungal activity. FLZ prevents fungus’ cytochrome P-450 sterol C-14 alpha-demethytion which causes an increase in the fungus’ 14 alpha-methyl sterol and loss of the normal fungal sterols, ultimately leading to fungistatic activity [11]. FLZ is characterized by its high molecular weight of 306.27 Da and poor water solubility [12,13]. Temporary mild-to-moderate increases in serum aminotransferase levels occur in up to 5% of FLZ-treated patients. However, clinically apparent hepatotoxicity due to FLZ is rare [14,15]. To reduce FLZ adverse effects, increase drug activity, and maintain drug delivery, many techniques have been used. Controlled release formulations for ophthalmic use have the advantage of reducing the systemic side effects. FLZ-loaded nanostructured lipid carriers [16], liposomes [17] and polymeric nanoparticles [18,19] were recently reported for the effective permeation of different antifungal drugs, hence achieving efficient treatment of complicated infections such as fungal keratitis.

In this work, an optimized FLZ-polymeric nanoparticle formulation was developed, characterized and their activity against a standard strain of *Candida albicans* was studied. The optimized nanoparticles were loaded into two different ophthalmic formulations. The rheological behaviour, in vitro drug release, safety, and ocular transport of the prepared ophthalmic formulations were studied to evaluate the effectiveness and safety of the prepared formulation.

## 2. Materials and Methods

### 2.1. Materials

Polycarpolactone (PCL), molecular weight 14,000, Polyethylene Glycol, molecular weight 400 (PEG 400), Tween^®^ 80, Dialysis tube, molecular weight cut off 10,000 and Fluorescein isothiocyanate (FITC)-dextran were purchased from Sigma-Aldrich (St. Louis, MO, USA). Polyvinyl alcohol (PVA) was obtained from Spectrum Chemicals & Laboratory Products (Gardena, CA, USA). Fluconazole (FLZ) was kindly gifted from Riyadh pharma (Riyadh, KSA). Acetone (99%) was purchased from PanReac AppliCem, ITW reagents (Barcelona, Spain). Hydroxypropyl methylcellulose (HPMC), molecular weight 86,000 g/mol, viscosity 4000 cp (2% solution) was purchased from Acros Organics (Morris Plains, NJ, USA). Poloxamer 407 was procured from Xi’an Lyphar Biotech Co., Ltd. (Xi’an, China). All other materials and solvents were of analytical grade. 

### 2.2. Design of Experiment

In this work, response surface methodology using the Box-Behnken design (BBD) was applied to correlate the dependent and independent variables that affect the development of FLZ nanoparticles. A three-factor at three-level design was utilized to generate 15 runs using the statistical package Statgraphics^®^ Centurion XV Software, Version 15.2.05 (StatPoint, Inc., Herndon, VA, USA). The percentage of PCL in the total polymeric matrix (X1), the addition rate of PCL (X2) and the stirring speed (X3) on the quality of the prepared nanoparticles were optimized. The dependent variables were particle size (Y1), zeta potential (Y2), and entrapment efficiency (Y3). The selected independent factors, their levels, and the studied responses are presented in Table 1. The composition of the 15 runs is listed in Table 2. The total polymeric matrix represents the amount of PCL and PEG in the formulation. 

### 2.3. Preparation of FLZ Nanoparticles

FLZ-loaded polymeric nanoparticles were prepared by the solvent evaporation method described elsewhere [20]. Briefly, a polymeric solution was prepared by dissolving a definite quantity of PCL in 30 mL acetone in a shaking water bath at 40 °C for 15 min. A known weight of FLZ (200 mg) was added to the prepared polymeric solution with continuous mixing until the complete dissolving of the drug. A hydrophilic phase (100 mL) was prepared by dissolving PEG in an aqueous solution of surfactants (a mixture of Tween^®^ 80 and polyvinyl alcohol in distilled water) [21]. A drug to polymeric matrix ratio of 1:10 was used. Under magnetic stirring, the hydrophobic phase was added dropwise to the hydrophilic phase employing the predetermined addition rate and mixing speed (rpm) assigned for each formulation in Table 2. The mixture was subjected to continuous overnight mixing until complete evaporation of the organic solvent and formation of a drug-loaded nanoparticles suspension. 

### 2.4. Characterization of the Prepared Polymeric Nanoparticles Formulations

#### 2.4.1. Particle Size, Polydispersity Index and Zeta Potential Measurements

The average particle size, polydispersity index (PDI), and the zeta potential value for the prepared 15 polymeric nanoparticle formulations were measured using a Malvern Zetasizer Nano ZSP (Malvern Panalytical Ltd., Malvern, UK). The prepared polymeric nanoparticles’ particle size and zeta potential were measured using the dynamic light-scattering with non-invasive backscatter optics and laser Doppler micro-electrophoresis, respectively. Data was analyzed using Malvern Zetasizer software version 7.12. Measurements were repeated in triplicate. 

#### 2.4.2. Entrapment Efficiency (EE) Measurement

The percentage of FLZ successfully entrapped in the polymeric nanoparticle formulations was estimated indirectly. Briefly, the known volume of the prepared polymeric nanoparticle formulation was placed in an Amicon^®^ falcon tube and centrifuged, using a 3-K30 Sigma Laboratory centrifuge (Ostrode, Germany), at 20,000 rpm for 60 min at 4 °C. The free drug supernatant was separated and analyzed spectrophotometrically at an λmax of 260 nm [22]. The drug, EE, was calculated using the following equation:(1)$$\mathrm{EE}\left(\%\right)=\frac{\left(\mathrm{Total}\;\mathrm{amount}\;\mathrm{of}\;\mathrm{FLZ}\;\mathrm{used}\;-\;\mathrm{Calculated}\;\mathrm{amount}\;\mathrm{of}\;\mathrm{free}\;\mathrm{FLZ}\;\mathrm{in}\;\mathrm{the}\;\mathrm{supernatant}\right)}{\left(\mathrm{Total}\;\mathrm{amount}\;\mathrm{of}\;\mathrm{FLZ}\;\mathrm{used}\right)}\times 100$$

### 2.5. Box-Behnken Design Statistical Analysis

Using Statgraphics software, the data obtained for the dependent variables (Y_1_, Y_2_, and Y_3_) were statistically analyzed at a *p*-value of 0.05 or less to determine the significant independent factors influencing each response. The optimum formulation that met the study’s objective was identified. This formulation was prepared and characterized for particle size (Y1), zeta potential (Y2), and entrapment efficiency (Y3), as described above. Moreover, the percent of FLZ loading into the optimized formulation was calculated according to the following equation:(2)$$\mathrm{Percent}\;\mathrm{of}\;\mathrm{FLZ}\;\mathrm{loading}=\frac{\mathrm{Total}\;\mathrm{amount}\;\mathrm{of}\;\mathrm{FLZ}\;\mathrm{entraped}\;\mathrm{in}\;\mathrm{the}\;\mathrm{nanoparticles}}{\mathrm{Amount}\;\mathrm{of}\;\mathrm{nanoparticles}\;}\times 100$$

### 2.6. Morphological Study 

To investigate the morphological properties of the prepared optimized nanoparticle formulation, transmission electron microscopy (TEM) images were taken using the TEM model JEM-1230 (JEOL, Tokyo, Japan). Briefly, a few drops of the optimized FLZ nanoparticle formulation were mounted on a carbon-coated grid and left for about 5 min to permit good adsorption of the nanoparticles on the carbon film. Filter paper was used to remove any excess liquid. Finally, a few drops of uranyl acetate (1%) were added, and the sample was investigated.

### 2.7. In-Vitro Antifungal Susceptibility Testing

The antifungal activity was tested against a standard strain of Candida albicans ATCC 76615, obtained from the microbiology laboratory, King Abdulaziz University Hospital, Jeddah, KSA.

The fungal stock cultures were cultured on Sabouraud dextrose agar plates and maintained at 37 °C. A loopful of these fungal cells was inoculated into 5 mL normal saline (85% NaCl) and incubated at 37 °C until achieving the desired concentration. Preliminary screening of the antifungal activities was conducted using an agar diffusion technique, as described previously [23]. Briefly, petri dishes (90 mm) were filled with 25-mL of Sabouraud dextrose agar containing 1 mL fungal culture (1 × 10^6^ CFU/mL) and the fungal strain was inoculated. A volume of 100 µL of the studied samples was added into holes of 12 mm diameter on the seeded agar plates. Dishes were preincubated for 2 h at 4 °C, then incubated for 48 h at 37 °C. Inhibitory activity was defined as the absence of fungal growth surrounding the holes. The inhibition zone was measured using a calliper. The antifungal activity of the prepared FLZ optimized nanoformulation was compared to a drug suspension sample (positive control) and drug-free nanoformulation (negative control).

### 2.8. Preparation of Ophthalmic Formulations

Poloxamer 407 was used in a concentration of 16% to prepare a thermosensitive in situ hydrogel (ISHG) formulation. The specified quantity of the polymer was dispersed in a known volume of the optimized drug-loaded polymeric nanoparticle formulation, previously cooled to 4°C, over a magnetic stirrer and the stirring process was continued until formation of the homogenous mixture without any lumps or precipitates. HPMC (0.5% *w*/*v*) was added to the polymeric solution as a viscosity modifier. Pure FLZ-loaded thermosensitive ISHG formulation was also prepared in cold deionized water. The prepared ISHG formulations were kept at 4°C until further characterization. 

Ophthalmic HPMC hydrogels were also prepared. In brief, the specified weight of HPMC (1% *w*/*v*) was added to a known volume of either the optimized FLZ polymeric nanoparticles or deionized water containing pure FLZ over a magnetic stirrer. The stirring process was continued until complete addition of the formulation ingredients and formation of homogenous mixture. The prepared hydrogels were stored at 4 °C until further characterization.

### 2.9. Rheological Behaviour 

The rheological behaviour of the prepared ophthalmic formulations (poloxamer 407-based ISGH and HPMC hydrogels) was studied using a Brookfield DV-II ultra-programmable cone and plate rheometer (Middleboro, MA, USA) at room temperature. Briefly, 0.5 g of each sample was placed inside the carefully closed plate. The spindle speed was increased gradually until reaching a torque in the range of 10 to 90%. Once the torque reached 90%, the speed was gradually reduced until reaching 10% torque. 

Viscosity values of the ISHG formulations were evaluated after inducing gelation by raising the temperature to 34 °C, while the HPMC HPMC-based hydrogels were measured directly without external stimuli.

### 2.10. In Vitro Drug Release 

The in vitro release of FLZ from the prepared ophthalmic ISHG and HPMC hydrogels formulations was studied using the dialysis bag method, as previously published [23]. Each formulation’s known weight (equivalent to 10 mg of the drug) was placed in a sealed dialysis bag with a molecular weight cut-off of 14 k Da, Sigma-Aldrich Inc. (St. Louis, MO, USA). A known volume of simulated tear fluid (STF; sodium chloride, sodium bicarbonate, and calcium chloride of 0.67% *w*/*v*, 0.2% *w*/*v* and 0.008% *w*/*v*, respectively, in deionized water, and the pH of the solution was adjusted to 7.4) was added to the studied formulation in a ratio of 25:7 (formulation: STF) to simulate the condition in the human eye [23]. The dialysis bag was placed in a glass bottle containing 250 mL of phosphate buffer of pH 7.4 and the bottles were kept at 32 °C under 100 rpm in a shaking water bath using model 1031 from the GFL Corporation (Burgwedel, Germany). Samples of 2 mL were taken from each bottle at specified time intervals with an instant replacement to maintain the sink condition. The concentration of FLZ in the collected samples was determined spectrophotometrically at 260 nm against a blank of the non-medicated formulation. The study was repeated three times. To investigate the FLZ release kinetics and the mechanism of drug release from the prepared ophthalmic ISHG and HPMC hydrogels formulations, the obtained results for the in vitro release data were fitted into several mathematical models such as: zero [24], first [24], Higuchi [25], and Korsmeyer–Peppas [26,27].

### 2.11. Rabbit Eye Irritation Test

To study the ocular irritation upon application of the prepared FLZ ophthalmic formulations, the rabbit eye irritation test was conducted according to the technique previously reported by Zhu et al. [28] and Ahmed et al. [23]. New Zealand white rabbits with an average weight of 2 kg were obtained from the animal housing of the Faculty of Pharmacy, KAU, Jeddah, KSA. The Ethics Committee of the Faculty of Pharmacy at King Abdulaziz University in Jeddah, Saudi Arabia, provided prior permission for this study (protocol approval date: June 2020, reference no. 1021441). The work was performed in accordance with the Declaration of Helsinki, the International Guiding Principle in Care and Use of Animals (DHEW publication no. NIH 80-23), and the Standards of Laboratory Animal Care (NIH distribution #85-23, reconsidered in 1985). Prior to conducting the experiment, rabbits were kept for about two weeks at 20 ± 1 °C with a 12/12-h dark/light cycle in naturally controlled rooms with free access to standard food and water. Twelve animals were randomly classified into three groups (n = 4). A formulation volume of 25 µL was installed into the lower conjunctival sac of the studied animal eyes using a micropipette. Group I (negative control group) were administered a non-medicated poloxamer 407-based ISHG formulation. Group II (test group) received the poloxamer 407-based ISHG formulation containing the optimized FLZ-polymeric nanoparticles. Group III (positive control group) was given the poloxamer 407-based ISHG formulation containing pure FLZ. Over the course of eight hours, the eyes of the treated animals were inspected, and the degree of eye irritation was estimated using the traditional Draize test [29].

### 2.12. Ocular Transport Study

A florescence laser microscope was used to investigate the permeation of the prepared optimized polymeric nanoparticles from the ISHG formulation and across the eye layers of healthy New Zealand white rabbits. Florescence isothiocyanate (FITC)-loaded polymeric nanoparticles were prepared and loaded into poloxamer 407-based ISHG using the same procedure mentioned above, except FITC replaced FLZ. Pure FITC-loaded poloxamer 407-based ISHG formulation was also prepared. Two groups (n = 6) of twelve animals weighing 1.5 to 2.5 kg each were used. The study was conducted using the same formulation volume (25 µL) and the technique described in the above section. Following the animals’ treatment, two rabbits from each group were sacrificed after 1, 2, and 4 h and the two eyes of each rabbit were removed instantaneously. The collected eyes were maintained in formalin. Longitudinal cut sections across the animal eyes were performed using microtome blades. Samples of paraffin wax were prepared and examined under a Zeiss Axio Observer D1 inverted fluorescence laser microscope (FLM) from Carl Zeiss AG (Oberkochen, Germany) using an excitation of 470/40 nm, a beam splitter of 495, and an emission of 525/50 nm.

## 3. Results and Discussion

In recent times, many new antifungal agents have been introduced to overcome fungal resistance. Hexamidine diisethionate [30] and benserazide hydrochloride [31] have been tested and reported to have activity against *Candida*. A combination of these antifungals, especially with FLZ, not only demonstrated a synergistic antifungal activity but also enhanced the broad spectrum antifungal effect [31]. Nanoparticles have been utilized for better ophthalmic penetration and ocular bioavailability [10]. To enhance the corneal permeation and antifungal activity of FLZ, polymeric nanoparticles were prepared using PCL coupled with PEG copolymer. The FDA has approved PCL for a number of biomedical uses especially when it is linked with PEG [32]. PEG is a highly biocompatible hydrophilic, non-ionic polymer that can be conjugated with nanoparticles in a variety of ways, including surface adsorption, covalent binding, and mixing while nanoparticles are being prepared [33]. In this work, the percentage of PCL in the total polymeric matrix, its addition rate and the stirring speed used to develop nanoparticles utilizing the solvent evaporation method were optimized to develop FLZ-loaded nanoparticles. We aimed to optimize the factors affecting the developed nanoparticles to minimize their size, maximizing their zeta potential and enhance the FLZ entrapment efficiency. Finally, we studied the efficiency of the optimized nanoparticles by conducting in vitro antifungal susceptibility testing, in vivo ocular irritation and ocular transport to the posterior segment.

### 3.1. Development of FLZ-Loaded Nanoparticles

As suggested by the BBD, 15 formulations of FLZ-loaded nanoparticles were successfully prepared and characterized. Table 2 shows the observed values of the studied responses. The mean particle sizes of the prepared nanoparticle formulations ranged from 114.3 ± 39.70 nm (F4) to 441.8 ± 181.4 nm (F13). The polydispersity index (PDI) ranged from 0.27 ± 0.03 to 0.38 ± 0.02, indicating a uniform size distribution of the prepared particles as previously mentioned for PDI value less than 0.7 [34]. The obtained zeta potential values varied from −5.67 ± 5.00 mV (F6) to −24.7 ± 7.94 mV (F3). All formulations displayed high drug entrapment efficiencies of more than 95%, which is expected to have a beneficial effect in the reduction of the administered dose during ophthalmic treatment. 

### 3.2. Optimization of the Polymeric Nanoparticles

Statistical analysis for the obtained results of Y1, Y2, and Y3 was performed using a two-way ANOVA followed by multiple regression analysis utilizing the Statgraphics^®^ software. Table 3 shows the statistical analysis of the independent factors on the studied responses Y1–Y3. The positive and negative signs of the estimated effect indicate a synergistic or antagonistic effect of this factor on the studied response, respectively. The F-ratio is used to indicate the presence of a location effect if the observed and expected averages are correlated and the F-ratio value was found to be greater than 1. Hence, the calculated *p*-value is used to determine whether a significant level exists. A significant effect is expected when there is a *p*-value less than 0.05 and different from zero.

The mathematical model for each response was generated and the following polynomial Equations (3)–(5) were obtained:Particle size (Y1) = −214.138 + 4.782 X1 − 21.596 X2 + 1.429 X3 + 0.164 X1X1 − 0.308 X1X2 − 0.023 X1X3 +3.388 X2X2 − 0.037 X2X3 − 0.0004 X3X3 (3)
Zeta potential (Y2) = 74.941 − 0.197 X1 − 0.184 X2 − 0.156 X3 − 0.009 X1X1 + 0.053 X1X2 + 0.0004 X1X3 −0.069 X2X2 − 0.0012 X2X3 + 0.0001 X3X3(4)
Entrapment efficiency (Y3) = 105.435 + 0.01 X1 − 0.047 X2 − 0.017 X3 − 0.002 X1X1 − 0.01 X1X2 +0.0004 X1X3 + 0.001 X2X2 + 0.0005 X2X3 + 0.0000002 X3X3(5)

#### 3.2.1. Impact of the Studied Factors on the Particle Size (Y1)

The release behaviour, cellular permeation, tissue distribution and the drug pharmacokinetics are significantly influenced by the nanoparticle size [35]. As shown in Table 2, the particle size ranged from 114.3 nm to 441.8 nm. The addition rate of PCL (X2) and the stirring speed (X3) were found to be the most important variables that affect the particle size, and both showed *p*-values less than 0.05, as indicated by the regression analysis (Table 3) and the Pareto chart (Figure 1).

It was found that there is a direct relationship between X2 and the particle size, whereas there is an inverse relationship between X3 and the particle size. Additionally, there is a strong direct correlation between the particle size and the quadratic terms of X1 and X2. The interaction term of X1X3 had a significant inverse effect on the particle size. The rest of the studied factors did not affect Y_1_. Figure 1a–d, which represent the Pareto chart and the three-dimensional response surface plots, respectively, illustrate that the PCL addition rate and the stirring speed significantly control the nanoparticles’ size (Y1). The increase in X2 from 5 to 15 mL/min resulted in an increase in the particle size of the produced nanoparticles from 234.6 nm (F5) to 429.3 nm (F7), from 150.4 nm (F10) to 179.4 nm (F12), from 228.9 nm (F9) to 404.9 nm (F2), and from 280.6 nm (F6) to 352.0 nm (F14) at the same levels of X1 and X3. This effect could be attributed to the increase in the polymer addition rate that results in an increase in the polymer (PCL) percentage/load and also results in enhancement in the tendency of particle aggregation, leading to aggregation of the small nanoparticles to form larger ones. Statistical analysis of the results (Table 3) also illustrates that there was an antagonistic effect of the stirring speed on the particle size. It was observed that increasing the speed from 400 rpm to 800 rpm resulted in a marked decrease in the particle size from 404.9 nm (F2) to 179.4 nm (F12), from 441.8 nm (F13) to 114.3 nm (F4), and from 228.9 nm (F9) to 150.4 nm (F10). The lower speed could diminish the shearing action which increases the viscosity of the dispersed system and resulting in larger nanoparticles [36].

#### 3.2.2. Impact of the Studied Factors on the Zeta Potential (Y2)

The net surface charge of the nanoparticles is mainly dependent on the functional groups of the formulation ingredients. The zeta potential is an important parameter that helps determine the nanoparticulate system’s stability since an electrostatic repulsion between particles of the same charges will prevent particle agglomeration and the stability of nanoparticle [22]. All the studied formulations demonstrated negative zeta values that ranged from −5.67 mV (F6) to −24.7 mV (F3). The percentage of PCL in the polymeric matrix was found to significantly affect the zeta potential value of the nanoparticles (Y2) as inferred from the Pareto charts (Figure 2a) and the three-dimensional response surface plots (Figure 2b–d). Statistical analysis of the results reveal that there was an indirect correlation between X1 and Y2. Increasing the percentage of PCL in the polymeric matrix from 20 to 60% results in a decrease in Y2 from −24.7 mV (F3) to −17.4 mV (F13), from −22.9 mV (F5) to −5.67 mV (F6), and from −16.5 mV (F11) to −14.8 mV (F4) at the same levels of X_2_ and X_3_. This finding could be attributed to increasing the hydrophobicity of the polymer matrix (i.e., PCL) which may result in increasing the negative charge due to the polyester nature of PCL [37,38,39].

#### 3.2.3. Impact of the Studied Factors on the EE % (Y3)

Various reports have demonstrated the effect of changing the polymeric matrix percent, addition rate and stirring speed on EE. As shown in Table 2, the EE % of FLZ varied slightly within a narrow range (95.0 to 99.99%) which indicates the successfulness of the employed technique in the development of nanoparticles having high drug entrapment. Additionally, Figure 3 represents the Pareto chart and the response surface plots for the effect of the studied variables on the drug entrapment efficiency (Y3) and indicate that this response was only affected by X1 at a *p*-value of 0.0111. As the percentage of X1 increased from 20 to 60%, the FLZ entrapment efficiency (Y3) decreased from 99.87% (F3) to 95.0% (F13), and from 99.99% (F7) to 95.0% (F14) when the same level of X2 and X3 were used. On the other hand, decreasing the addition rate at a fixed stirring speed could result in enhancing the encapsulation efficiency (99.9%, 99.0% for F5 and F6, respectively). Our results demonstrate that in order to achieve high EE, optimised levels of three factors is needed. 

#### 3.2.4. Validation of the Optimized FLZ-Loaded Nanoparticles

An optimized nanoparticle formulation was proposed, prepared and characterized to confirm the practicability of the optimization technique. Combining X1, X2 and X3 at 20%, 6.27 mL/min and 400 rpm, respectively, resulted in maximizing the desirability function of the factors over the design space. This formulation showed a size of 145.5 nm as depicted in Figure 4. No marked residuals were obtained when the observed and predicted response values were compared which indicates the effectiveness of the BBD design for optimization of the prepared nanoparticle. Table 4 displays the results of the observed and expected values. Moreover, the prepared optimized formulation demonstrated a drug loading of 8.9%.

### 3.3. Morphological Study of the Optimized Polymeric Nanoparticles

Results of the TEM images for the optimized nanoparticles reveal that the prepared drug-loaded nanoparticles were spherical in shape and about 200 nm in size. This finding is consistent with a previous report that demonstrated the spherical shape of diosgenin-loaded PCL-Pluronic nanoparticles [40]. Figure 5 reveals the presence of a dark thin layer around the particles which could be attributed to the deposition of PEG chains on the polymeric nanoparticle surface as previously described by Zanetti-Ramos for pegylated polyurethane nanoparticles [41]. The slight difference in size between the dynamic light-scattering technique and TEM could be attributed to the polydispersity index of the sample as previously mentioned during the characterization of polyurethane nanoparticles by dynamic light scattering and atomic force microscopy techniques [42]. 

### 3.4. In Vitro Antifungal Susceptibility Testing

An agar diffusion assay was used to study the in vitro antifungal behavior of the prepared formulations using representative standard strains of fungus. Antifungal activities of the tested formulations against *Candida albicans* ATCC 76615 showed varying results (Figure 6). No inhibition zone was noticed with a drug-free nanoformulation (negative control). A medium inhibition zone of 15 mm was observed with the plain drug. A marked inhibition zone of about 28 mm was observed with the optimized nanoformulation. The rationale for this finding is the improvement in drug diffusion from the drug-loaded PCL nanoparticles which enhanced the drug antifungal activity as previously mentioned for FLZ-loaded alginate/chitosan-based polymeric nanoparticles against *Candida albicans* [43].

### 3.5. Rheological Behaviour of the Prepared Ophthalmic Formulations 

As the shear stress was increased at 34 °C (gelation external stimuli), the prepared thermosensitive ISHG formulations show a marked decrease in the apparent viscosity. The same behavior was noticed with the prepared HPMC hydrogel formulations; however, because these formulations (HPMC hydrogels) are viscous liquids that cannot change when exposed to external stimuli, such as temperature elevation, there was no need for induction of gelation. Accordingly, all the studied ophthalmic formulations were found to be of a non-Newtonian pseudoplastic property. Accordingly, all the tested formulations are expected to have a shear-thinning flow behavior by which the sample’s viscosity is decreased at a higher shear rate or angular velocity “rpm” (figure not shown). A previous study indicated that the sol-gel transition temperature for the poloxamer 407 thermo-gelling system was prepared using 15–20% *w*/*w* polymer in cold water and is in the range of 21.9 to 38.5 °C [44]. The sol-gel transition temperature of 18% *w*/*w* poloxamer 407 aqueous solution is decreased to 22–25 °C upon addition of some additives such as HPMC (0.5–1% *w*/*w*), polyvinyl pyrrolidone (4% *w*/*w*), xanthan gum (0.2% *w*/*w*) or carrageenan (0.2% *w*/*w*). In this work, the viscosity measurements were conducted at 34 °C for two reasons. First, to ensure complete sol-gel transition of the poloxamer-based ISHG formulation. Second, to mimic the physiological eye condition since the eye has an average surface temperature of 34.51 ± 0.82 °C [45]. The results of the viscosity measurements indicate higher viscosity values of the ISHG formulations when compared to the HPMC hydrogels. Poloxamer-based ISHG reveal viscosity values of 34.9 ± 1.4 and 36.7 ± 2.2 (cP × 10^3^) for nanoparticles and pure drug-loaded formulations, respectively. The HPMC hydrogel formulations show viscosity values of 1332 ± 65 and 1216 ± 57 (cP) for nanoparticles and pure drug-loaded formulations, respectively. These results indicate that there was no significant difference in the viscosity between the nanoparticles and drug-loaded formulations. At rest, pseudoplastic systems usually have a disarranged molecular structure, with the molecules of the polymer coiled up in their globular form. As the shearing force is raised, the polymer chains begin to untangle, and the molecules start to align their axis in the direction of flow. It is anticipated that addition of HPMC to the ISHG formulations will improve this feature and increase the system viscosity.

### 3.6. In Vitro Release Study and Kinetic Treatment of the Data

The studied ophthalmic preparations demonstrate a prolonged drug release pattern that lasted for eight hours (Figure 7). Loading the pure drug or the optimized nanoparticles into ISHG and HPMC hydrogel formulations extended the drug release due to the three-dimensional polymeric network which allows drug release only by diffusion from these formulations [46]. It was previously mentioned that hydrogel delivery systems act as a reservoir or a matrix device that control the drug release. In the former (reservoir system), the solid drug particles are present in the core and the system ensures a constant rate of drug release, while in the later (matrix system), the drug is uniformly distributed in the hydrogel matrix as solid particles [47]. A superior drug release profile was observed from the prepared polymeric nanoparticles when compared to the pure drug-loaded formulation, as depicted in Figure 7. A possible explanation for this behaviour is the existence of the drug, in the prepared polymeric nanoparticles, in the colloidal range which facilitates diffusion from the hydrogel reservoir or matrix. On the contrary, the pure drug particles exist in the coarse range which hinder their diffusion. A high cumulative percent of FLZ release (at 8 h) was observed from the nanoparticle-loaded ISHG (84.89 ± 17.5%) than from the corresponding pure drug-loaded ISHG (40.28 ± 4.2%). On the other hand, the prepared HPMC hydrogel demonstrated a cumulative percent of drug release of 65.65 ± 13.5% and 31.5 ± 3.2% from the formulations loaded with nanoparticles and pure drug, respectively. It is noteworthy to mention that the prepared ophthalmic formulations (ISHG and HPMC hydrogel) are expected to be applied right before bed since both might make the patient’s vision sightly blurry due to their viscous nature. Accordingly, eight hours of drug release was studied which is equivalent to the average sleep time for a young adult (7–9 h) [48]. 

According to the value of the maximum correlation coefficient (R) for the studied kinetic models, the release of FLZ from the prepared formulations was found to follow zero-order kinetics. The drug release was independent of the amount of drug released at various time points. These results are consistent with earlier research which indicated that the release of itraconazole from polymeric micelles loaded into an in situ ocular gel was found to follow zero-order kinetics [49]. Moreover, another study reported a zero-order release pattern for ketoconazole from a carbopol-based in situ hydrogel [23].

It is worth mentioning that biocompatibility and biodegradability are key parameters that affect the choice of the polymer used to develop the formulation. PCL is a biocompatible and biodegradable polymer that was used in ocular drug delivery [9]. Due to its ability to extend the drug release pattern, it has been exploited in ocular drug delivery technologies [50,51]. Moreover, it can produce a porous implant surface that can tune the drug release for the ocular implants [8]. Incorporating PEG, a hydrophilic biocompatible polymer, in the nanoparticle formulation increases the mucoadhesive properties and improves the drug ocular delivery [33]. Poloxamer 407 is a well-known hydrophilic, gelling stimuli-responsive polymer that is used in ocular delivery to extend drug release time [52,53]. HPMC can help increase the mucoadhesive properties and extend the contact time with the ocular tissue. Mansour et al. has reported optimum ciprofloxacin HCl release with enhanced mucoadhesive property using an in situ forming gel composed of poloxamer 407 and HPMC [54]. Accordingly, FLZ-loaded PCL nanoparticles conjugated with PEG is a promising drug delivery cargo particularly when loaded into a Pluronic-based thermosensitive in situ gel, however, their ocular irritation and transport need to be confirmed.

### 3.7. Rabbit Eye Irritation Test

Most of the currently available marketed antifungal drugs have drawbacks, including poor absorption and restricted ocular penetration, particularly in patients with deep-seated infections [10]. Amphotericin B is a well-known drug markedly used by practicing physicians to treat fungal eye infections, but there are many disadvantages in using this drug [55]. Natamycin, another topical antifungal agent, is characterized by its poor aqueous solubility that restricts its entry into the anterior chamber and deep corneal layers, limiting its usage as a monotherapy to superficial keratitis [56]. The potential toxicity and possible drug interactions limit the oral use of ketoconazole, although it has a better safety profile than amphotericin B and it possesses broad-spectrum activity [57]. Filler et al. studied the activity of FLZ on the long-term therapy of endophthalmitis in rabbits with disseminated candidiasis after intravenous drug administration and reported a loss of FLZ activity after 24 h [58]. In this work, we have developed a FLZ in situ gel formulation suitable for ocular application. This ophthalmic route is characterized by a low incidence of systemic side effects due to the passage of a slight amount of the administered formulation through the retinal-blood barrier. Therefore, no drug systemic side effects are expected following treatment with this formulation. 

Our research group has previously developed a carbopol-based ophthalmic formulation containing ketoconazole trans-ethosomal vesicles to treat fungal eye infections. The developed formulation was able to treat deep fungal eye infections without causing irritation to the cornea [23]. In this study, the ocular irritation and transport across the eye tissue of the developed formulations were also studied to investigate their safety and efficacy. New Zealand white rabbits were used to investigate the ocular irritation. The results revealed that the prepared poloxamer 407-based ISHG formulations were non-irritants and could be tolerated by the eye. There were no macroscopic signs of iris redness, edema, aberrant secretions, congestion, corneal opacity, bleeding, or significant damage. Comparing the animal eyes to the control, all were normal (Figure 8). As a result, the developed thermosensitive ISHG formulations received a total score of zero in the Draize test’s traditional irritation assessment, making these formulations safe to administer to the eyes. A related discovery was made after the use of ophthalmic in situ gel formulations loaded with pure FLZ, FLZ-hydroxypropyl-beta-cyclodextrin complex-loaded noisome vesicles and Eudragit-based nanoparticles in the treatment of eyes infection [59,60]. In the next section, corneal structure and histopathology of the eye tissues will be investigated.

### 3.8. Ocular Transport Study

It was previously mentioned that size reduction facilitates easy drug uptake by cells, allowing for effective drug delivery [61]. To confirm this concept, the transport of the optimized polymeric nanoparticles from the prepared poloxamer 407-based ISHG across the eye tissue was investigated. Polymeric nanoparticles loaded with FITC were used. Following treatment of the rabbit’s eyes with the nanoparticle formulation loaded with FITC, florescence laser microscopic pictures in the cornea (anterior segment) and retina (posterior segment) were taken and the obtained results are represented in Figure 9. Marked differences in the florescence intensity were observed in the anterior segment among the studied groups over all time points. Easy diffusion of the nanoparticles loaded with FITC resulted in a high florescence intensity after 1, 2 and 4 h. Low florescence intensity was detected in the group treated with the pure FITC formulation. Results for the posterior part of the eye (retina) was consistent with that of the cornea. Poor transport of the pure FITC to the posterior part resulted in a florescence that could hardly be observed. This finding is mainly attributed to distribution of the pure FITC in the ISHG matrix as coarse particles, since this dyne is characterized by its poor aqueous solubility (less than 0.1 mg/mL in water), which hinders its transport across the ocular tissues by the effect of the tears through quick washing of the dye and by eye-blinking action as previously reported [23]. It must be mentioned that the prolonged ocular residence of the ISHG formulation loaded with pure FITC formulation may result in the transport of a limited amount of the dye particles to the posterior part. Löscher et al. previously mentioned that following corneal treatment, the drug may reach the posterior eye segment from the anterior part by absorption into the conjunctiva which is a mucous membrane that begins at the edge of the cornea and lines the inside surface of the eyelids and sclera. The absorption process may be achieved by diffusion through the sclera or cornea or via systemic circulation [1]. Drug-loaded nanoparticles are expected to follow the same pathway. 

Accordingly, the prepared polymeric nanoparticles were found to be effective in the transport of their payload to the posterior eye segment to treat the deep fungal infection especially when these nanoparticles are loaded into a system that can prolong their contact with the eye, such as the ISHG.

## 4. Conclusions

In this work, the Box-Behnken optimization design was utilized to develop FLZ-loaded PCL nanoparticles. The optimized nanoparticles demonstrated a high drug entrapment efficiency, a negatively charged particle surface and a spherical shape. Higher antifungal activity against *Candida albicans* was noticed from the FLZ-loaded nanoparticles when compared to the pure drug. Thermosensitive ISHG and HPMC hydrogel ophthalmic formulations loaded with either the polymeric nanoparticles or the pure drug were developed and illustrate pseudoplastic flow properties. The ophthalmic formulations exhibit a controlled in vitro drug release profile with a superior drug release behaviour from the polymeric nanoparticles when compared to the corresponding pure drug-loaded formulation. Upon application of the thermosensitive ISHG formulations into the eyes of New Zealand white rabbits, the nanoparticles-loaded ISHG formulation was able to efficiently transport the drug to the posterior eye segment when compared to the corresponding pure drug-loaded ISHG formulation, although both formulations were tolerated well by the rabbits’ eyes. More studies are required to investigate the stability, ocular pharmacokinetics and the antifungal activities on fungal-infected mice models before considering the prepared poloxamer 407-based ISHG-loaded with FLZ PCL nanoparticles as a promising drug delivery system in the treatment of deep ocular fungal infections.

## Figures and Tables

**Figure 1 polymers-15-00209-f001:**
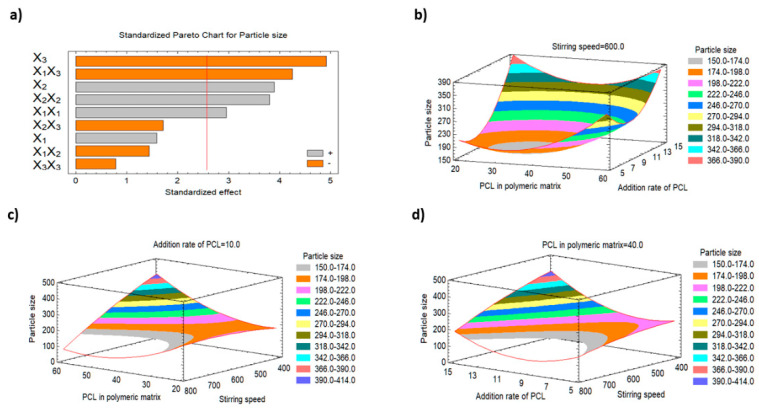
Pareto chart and the three-dimensional response surface plots representing the effect of changing the variables’ levels on the particle size (Y1). (**a**) Standardized Pareto chart for particle size. (**b**) The three-dimensional response surface plot of the effect of stirring speed on particle size (Y1). (**c**) The three-dimensional response surface plot of the effect of addition rate of PCL on particle size (Y_1_). (**d**) The three-dimensional response surface plot of the effect of PCL in polymeric matrix on particle size (Y1).

**Figure 2 polymers-15-00209-f002:**
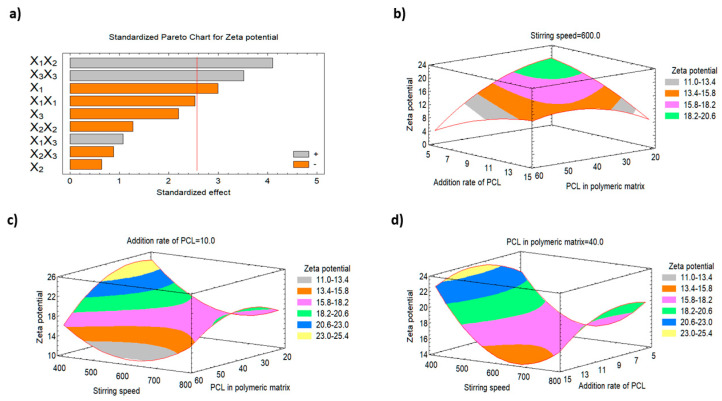
Pareto chart and the three-dimensional response surface plots representing the effect of changing the variables levels on the zeta potential (Y2). (**a**) Standardized Pareto chart for zeta potential. (**b**) The three-dimensional response surface plot of the effect of stirring speed on zeta potential (Y2). (**c**) The three-dimensional response surface plot of the effect of addition rate of PCL on zeta potential (Y2). (**d**) The three-dimensional response surface plot of the effect of PCL in polymeric matrix on zeta potential (Y2).

**Figure 3 polymers-15-00209-f003:**
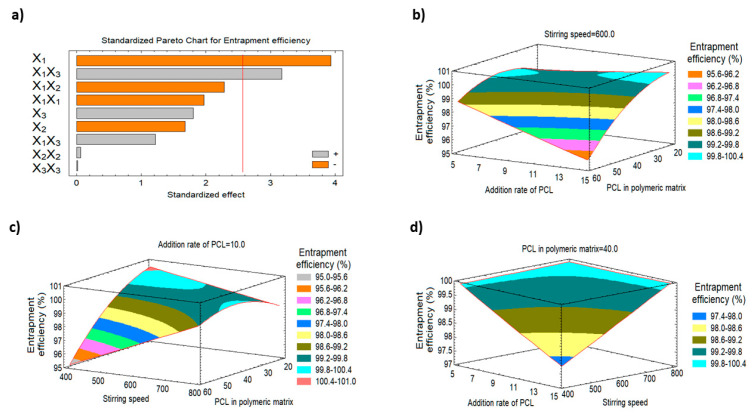
Pareto chart and the three-dimensional response surface plots representing the effect of changing the variables levels on the entrapment efficiency (Y3). (**a**) Standardized Pareto chart for entrapment efficiency. (**b**) The three-dimensional response surface plot of the effect of stirring speed on entrapment efficiency (Y3). (**c**) The three-dimensional response surface plot of the effect of addition rate of PCL on entrapment efficiency (Y3). (**d**) The three-dimensional response surface plot of the effect of PCL in polymeric matrix on entrapment efficiency (Y3).

**Figure 4 polymers-15-00209-f004:**
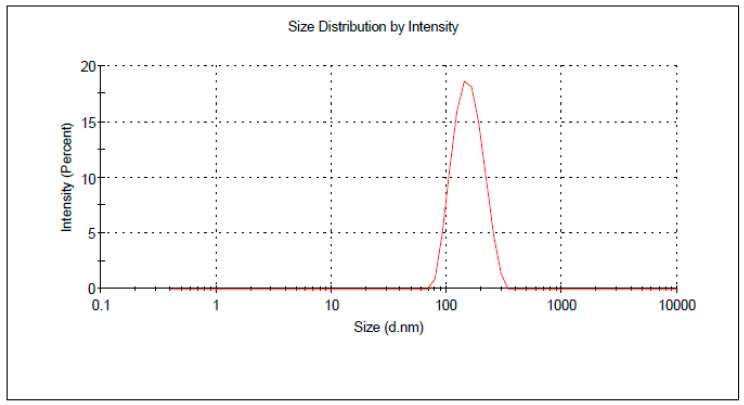
Particle size distribution of the optimized formulation.

**Figure 5 polymers-15-00209-f005:**
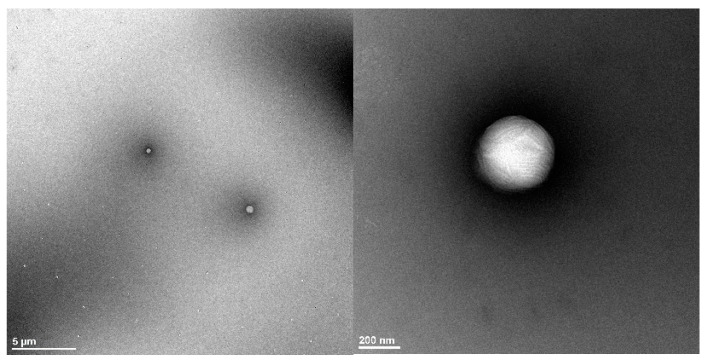
TEM images of the optimized FLZ nanoparticle formulation.

**Figure 6 polymers-15-00209-f006:**
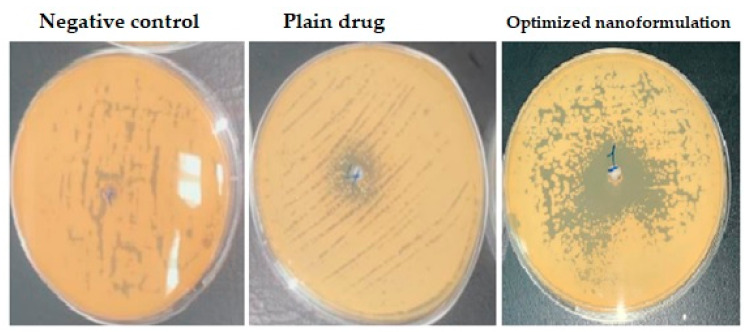
In vitro antifungal susceptibility testing.

**Figure 7 polymers-15-00209-f007:**
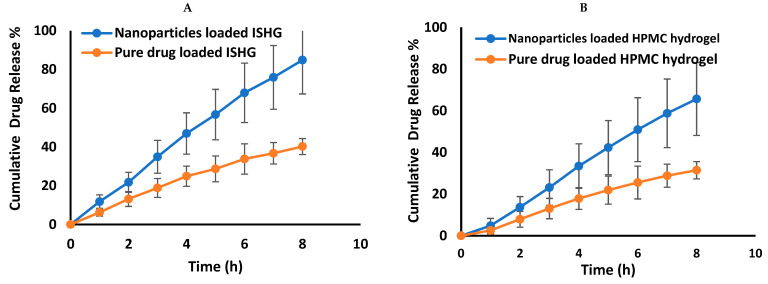
In-vitro FLZ release from nanoparticles and pure drug-loaded ISHG (**A**) and HPMC hydrogels (**B**).

**Figure 8 polymers-15-00209-f008:**
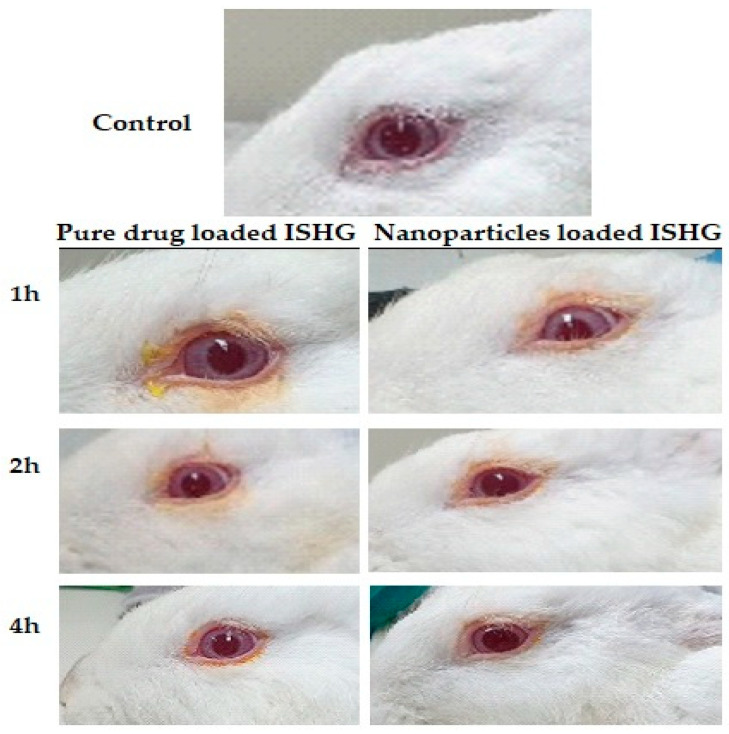
Macroscopic examination of the eye treated with the prepared ophthalmic formulations.

**Figure 9 polymers-15-00209-f009:**
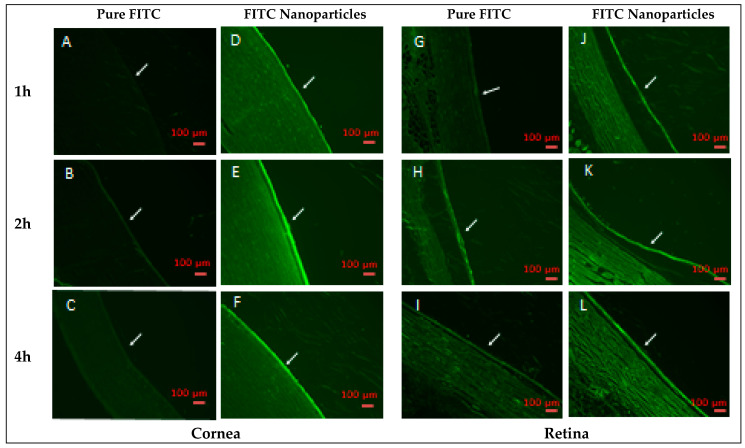
Images of a florescence laser microscope show eye sections on New Zealand white rabbits’ eyes after treatment with the ISHG formulation containing FITC-nanoparticles and the ISHG formulation loaded with pure FITC after 1, 2, and 4 h. Cornea sections (**A**–**F**) and retina sections (**G**–**L**) of rabbits’ eyes from both groups (arrows point to fluorescence). Abbreviations: FITC, florescence isothiocyanate; ISHG, in situ hydrogel.

**Table 1 polymers-15-00209-t001:** Independent and dependent variables and their levels utilized in Box-Behnken design.

Independent Variables	Low Level (−1)	Medium Level (0)	High Level (+1)	Responses	Units
PCL in the polymeric matrix, % (X1)	20	40	60	Particle size	nm
Addition rate of PCL, mL/min (X2)	5	10	15	Zeta potential	mV
Stirring speed, rpm (X3)	400	600	800	Entrapment efficiency	%

**Table 2 polymers-15-00209-t002:** Composition matrix and the observed values of the prepared FLZ-nanoparticles based on the Box-Behnken design.

Formulation Code	Independent Factors			Dependent Factors		
	X_1_	X_2_	X_3_	Y_1_ (nm)	Y_2_ (mV)	Y_3_ (%)
F1	40	10	600	169.4 ± 50.58	−17.1 ± 5.50	99.21 ± 5%
F2	40	15	400	404.9 ± 50.0	−23.8 ± 6.35	98.64 ± 5%
F3	20	10	400	147.7 ± 52.20	−24.7 ± 7.94	99.87 ± 5%
F4	60	10	800	114.3 ± 39.70	−14.8 ± 7.39	99.87 ± 5%
F5	20	5	600	234.8 ± 122.3	−22.9 ± 7.79	99.9 ± 5%
F6	60	5	600	280.6 ± 134.7	−5.67 ± 5.00	99.0 ± 5%
F7	20	15	600	429.3 ± 169.0	−7.36 ± 4.07	99.99 ± 5%
F8	40	10	600	191.3 ± 61.3	−17.7 ± 7.00	99.0 ± 5%
F9	40	5	400	228.9 ± 73.18	−19.0 ± 8.00	99.9 ± 5%
F10	40	5	800	150.4 ± 33.23	−18.6 ± 6.63	99.06 ± 5%
F11	20	10	800	184.1 ± 62.3	−16.5 ± 4.84	99.06 ± 5%
F12	40	15	800	179.4 ± 120.1	−18.8 ± 3.69	99.99 ± 5%
F13	60	10	400	441.8 ± 181.4	−17.4 ± 4.41	95.0 ± 6%
F14	60	15	600	325.7 ± 159.5	−11.5 ± 5.58	95.0 ± 6%
F15	40	10	600	160.3 ± 35.02	−16.2 ± 7.24	99.87 ± 5%

Notes: Values are considered as average ± standard deviation (n = 3). Abbreviations: X1, PCL % in polymeric matrix; X2, the addition rate of PCL in mL/min; X3, the stirring speed in rpm; Y1, particle size; Y2, zeta potential; Y3, entrapment efficiency.

**Table 3 polymers-15-00209-t003:** Statistical analysis with the estimated effects of factors, F-ratio, and associated *p*-values for the prepared FLZ-loaded nanoparticles.

Factors	Particle Size (Y1), nm			Zeta Potential (Y2), mV			Entrapment Efficiency (Y3), %		
	Estimate	F-Ratio	*p*-Value	Estimate	F-Ratio	*p*-Value	Estimate	F-Ratio	*p*-Value
X1	48.25	2.54	0.1718	−5.523	8.99	0.0302 *	−2.488	15.44	0.0111 *
X2	117.775	15.14	0.0115 *	−1.178	0.41	0.5508	−1.061	2.81	0.1547
X3	−148.775	24.16	0.0044 *	−4.05	4.83	0.0792	1.144	3.26	0.1306
X1X1	131.533	8.72	0.0318 *	−6.853	6.39	0.0527	−1.835	3.87	0.1061
X1X2	−61.65	2.07	0.2094	10.685	16.82	0.0093 *	−2.045	5.22	0.0712
X1X3	−181.95	18.07	0.0081 *	2.8	1.16	0.3316	2.844	10.08	0.0247 *
X2X2	169.383	14.45	0.0126 *	−3.453	1.62	0.2589	0.056	0.00	0.9548
X2X3	−73.5	2.95	0.1466	−2.3	0.78	0.4177	1.094	1.49	0.2764
X3X3	−34.917	0.61	0.4687	9.533	12.36	0.0170*	0.017	0.00	0.9862
R^2^	94.63			91.67			89.42		
Adj-R^2^	84.97			76.69			70.37		

Note: * Significant effect of a factor on individual response. Abbreviations: X1, PCL % in polymeric matrix; X2, the addition rate of PCL in mL/min; X3, the stirring speed in rpm; Y1, particle size; Y2, zeta potential; Y3, entrapment efficiency; X1X2, X1X3, and X2X3 are the interaction terms between the factors; X1X1, X2X2, and X3X3 are the quadratic terms between the factors; R^2^ = R-squared; Adj-R^2^ = adjusted R^2^.

**Table 4 polymers-15-00209-t004:** The optimum levels for the independent variables and values of the studied dependent variables.

Independent Variables	Optimum	Dependent Variables	Predicted Values	Observed Values	Residuals
PCL in polymeric matrix, % (X1)	20	Particle size (nm)	134.05	145.5	11.45
Addition rate of PCL, mL/min (X2)	6.27	Zeta potential (mV)	−27.14	−29.23	2.09
Stirring speed, rpm (X3)	400	Entrapment efficiency (%)	100.6	98.2	−2.4

## Data Availability

Not applicable.

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
