# Peer review of "Enhancing the Antifungal Activity and Ophthalmic Transport of Fluconazole from PEGylated Polycaprolactone Loaded Nanoparticles"

_polymers, 2022, doi:10.3390/polym15010209_

Round 1

Reviewer 1 Report

The authors developed a drug delivery system of FLZ-loaded poly (ε-caprolactone) nanoparticles and investigated the antifungal activity, ocular irritation, and transport of the drug delivery system using a rabbits’ eye model. The questions are as following.

1. Please provide the graphical results of the optimized nanoparticles' particle size and zeta potential  obtained by dynamic light-scattering method. 

2. The zata potential values are positive or negative? The values in section 3.1 are not consistent with that of table2.

3. Please provide the relative images of in vitro antifungal susceptibility testing.

4. As shown in figure 6, the cumulative drug release from the FLZ-loaded nanoparticles is higher than that of pure drug. What's the reason for that? How many drug was loaded into the nanoparticles? That is to say, please clarify  the drug loading of optimized nanoparticle formulation.

5. It's a little blurry for Figure 7 and Figure 8. Please provide the high resolution images.

6. For Figure 8, the cornea's thickness seems different from each image. Please clarify the reason and provide the high resolution images including all lays of the  cornea and retina.

What's the potential pathway for nanoparticles to enter into the posterior eye segment?

Author Response

Reviewer 1

The authors developed a drug delivery system of FLZ-loaded poly (ε-caprolactone) nanoparticles and investigated the antifungal activity, ocular irritation, and transport of the drug delivery system using a rabbits’ eye model. The questions are as following.

  1. Please provide the graphical results of the optimized nanoparticles' particle size and zeta potential  obtained by dynamic light-scattering method. 

Reply

The required results for particle size and zeta potential value obtained by dynamic light-scattering for the optimized formulation are submitted as a supplementary data.

  1. The zata potential values are positive or negative? The values in section 3.1 are not consistent with that of table2.

Reply

The prepared polymeric nanoparticles showed a negative zeta potential values. Values depicted in table 1 is the correct ones. The text has been corrected. We apologize for this typo mistake.

  1. Please provide the relative images of in vitro antifungal susceptibility testing.

Reply

The requested images for in vitro antifungal activity are submitted as a supplementary data.

  1. As shown in figure 6, the cumulative drug release from the FLZ-loaded nanoparticles is higher than that of pure drug. What's the reason for that? How many drug was loaded into the nanoparticles? That is to say, please clarify  the drug loading of optimized nanoparticle formulation.

Reply

  • The existence of the drug in the form of a polymeric nanoparticles in the colloidal range facilitated its diffusion from the prepared ophthalmic formulations. On the other hand, the pure drug is expected to be existed in the coarse range which hinders its diffusion. This explanation is illustrated in the modified manuscript (section 3.6).
  • Known weight (equivalent to 10 mg drug) from the prepared ophthalmic ISHG and HPMC hydrogels formulations was placed in a glass bottle containing 250 mL of phosphate buffer of pH 7.4. This information is illustrated in the modified manuscript (section 2.10).
  • A drug to polymer ratio of 1:10 was used and the % drug loading of the optimized nanoparticles formulation was found to be 8.9%. This information is illustrated in the modified manuscript (section 3.2.4.).

  1. It's a little blurry for Figure 7 and Figure 8. Please provide the high resolution images.

Reply

  • High resolution images have been provided.
  • The individual components of each figure have been submitted as a supplemanatry data .

  1. For Figure 8, the cornea's thickness seems different from each image. Please clarify the reason and provide the high resolution images including all lays of the  cornea and retina.

Reply

  • More clear figure has been uploaded in the revised manuscript.
  • The individual components for figure 8 have been submitted as supplementary data.
  • The reason for the difference in the cornea’s thickness may be attributed to individual variations between animals or due to the difference in cut section taken or handling procedure.

What's the potential pathway for nanoparticles to enter into the posterior eye segment?

Reply

The potential pathway for nanoparticles to reach the posterior eye segment has been clarified in the revised manuscript (section 3.8.).

Reviewer 2 Report

Dears authors

Analysis by paper partitions:

1 - Introduction: the content and drafting of the general part needs to be reformed, review the syntax of the topic

2- Discussion:

deepen the discussion on the use of new ophthalmic drugs against MDR candida, optionally use these suggested literature items and discuss, expanding the discussion by at least 3 lines:

PMID: 32452982 ; PMID: 34815665 ; PMID: 34829196 

3 - Check the bibliographic entries throughout the text, some of which are non-compliant.

4 - Review English grammar and in particular applied scientific English: in particular verb tenses and syntax in discussion.

Author Response

Reviewer 2

1 - Introduction: the content and drafting of the general part needs to be reformed, review the syntax of the topic.

Reply

The introduction section has been reformed and the following sequence has been included in the revised manuscript:

  • Challenges with the process of ocular drug delivery.
  • Nanaotechnology based drug delivery system with focusing on the polymeric particles.
  • Antifungal agents with focusing on Fluconazole.

 2- Discussion:

deepen the discussion on the use of new ophthalmic drugs against MDR candida, optionally use these suggested literature items and discuss, expanding the discussion by at least 3 lines:

PMID: 32452982 ; PMID: 34815665 ; PMID: 34829196 

Reply

The discussion section has been modified to include the use of new ophthalmic drugs and the role of nanoparticles in ocular delivery of antifungal agents (3. Results and Discussion, first paragraph).

3 - Check the bibliographic entries throughout the text, some of which are non-compliant.

Reply

All cited references have been checked and update and the non-compliant ones have been removed in particular the results and discussion part (section 3.7.).   

4 - Review English grammar and in particular applied scientific English: in particular verb tenses and syntax in discussion.

Reply

The discussion section has been reviewed for grammar in particular verb tenses.